# Age-Stage, Two-Sex Life Tables of *Megalurothrips usitatus* (Bagnall) and *Frankliniella intonsa* (Trybom) on Different Bean Pods Under Laboratory Conditions: Implications for Their Competitive Interactions

**DOI:** 10.3390/insects15121003

**Published:** 2024-12-18

**Authors:** Mengni Li, Zhengke Peng, Chaosong Guo, Yong Xiao, Fei Yin, Haibin Yuan, Zhenyu Li, Myron P. Zalucki

**Affiliations:** 1Department of Plant Protection, Jilin Agricultural University, Changchun 130118, China; lininim@163.com; 2Institute of Plant Protection, Guangdong Academy of Agricultural Sciences, Key Laboratory of Green Prevention and Control on Fruits and Vegetables in South China Ministry of Agriculture and Rural Affairs, Guangdong Provincial Key Laboratory of High Technology for Plant Protection, Guangzhou 510640, China; zkpeng0827@163.com (Z.P.); guo.cs@foxmail.com (C.G.); xiaoyong@gdaas.cn (Y.X.); feier0808@163.com (F.Y.); 3Guangdong Provincial Key Laboratory of Insect Developmental Biology and Applied Technology, Institute of Insect Science and Technology, School of Life Sciences, South China Normal University, Guangzhou 510631, China; 4School of the Environment, The University of Queensland, St Lucia, QLD 4072, Australia; m.zalucki@uq.edu.au; 5Shandong Engineering Research Center for Environment-Friendly Agricultural Pest Management, College of Plant Health and Medicine, Qingdao Agricultural University, Qingdao 266109, China

**Keywords:** thrips, two-sex life table, leguminous crops, competitive interactions, cut surfaces

## Abstract

This paper reports differences in the development of *Megalurothrips usitatus* (Bagnall) and *Frankliniella intonsa* (Trybom) on three legume plants, providing a theoretical basis for interspecific competition between these two species. The results showed that *M. usitatus* were more competitive on snap bean, *F. intonsa* were more competitive on cowpea, and both thrips were less adapted on green bean. Moreover, cut surfaces of cowpea did influence the performance of *M. usitatus*. The fecundity of *M. usitatus* on end-wrapped cut cowpea decreased compared to uncovered ones. Uncut cowpea pods are closer to the actual living conditions of *M. usitatus* in the field and it provides a new perspective for the study of thrips’ life table parameters in a laboratory.

## 1. Introduction

Thrips (Thysanoptera) are important agricultural pests worldwide, causing harm to leguminous, cucurbitaceous, solanaceous, and liliaceous crops [1,2,3]. *Megalurothrips usitatus* (Bagnall) has become a major agricultural pest, damaging crops such as *Luffa cylindrica* (L.), *Raphanus sativus* L., *Solanum melongena* L., and various flowers, with major damage on legumes, especially cowpea [4]. Similarly, *Frankliniella intonsa* (Trybom) is a polyphagous pest, attacking similar hosts of *M. usitatus*, including cowpea, and seriously affecting its production in China. It has previously been reported that *M. usitatus* is gradually replacing *F. intonsa* as the dominant pest thrips in cowpea [5].

A life table analysis is a critical tool for studying insect population dynamics and can be used to evaluate the effects of different host plants on pest development, survival, reproduction, and population growth [6]. Traditional age-specific life tables consider only females, ignoring males and failing to differentiate between instar stages. In contrast, age-stage, two-sex life tables can be used to study the impact of sex ratios on population survival, development, and reproduction, accurately describing instar differentiation [7].

Differences in food sources can affect insect size (body length and weight), survival rate, growth, development, and fecundity [8,9]. Previous research on host plants of *M. usitatus* and *F. intonsa* have primarily focused on cowpea (*Vigna unguiculata* L. var. Chunqiu Wujiadou) but studies show that *M. usitatus* can complete its life history on snap bean (*Phaseolus vulgaris* L. var. Yonglong 3) with higher survival rates and fecundity than on cowpea [10,11]. It remains unclear whether *F. intonsa*, in the same niche as *M. usitatus*, can also live on snap bean. Additionally, it is uncertain whether green bean (*Phaseolus vulgaris* L. var. Linghangzhe), a different variety of the same species as snap bean, is suitable to *M. usitatus*.

Previous laboratory life table studies have used pod segments [10,11,12]. In our experiments, we observed that adult thrips preferred to feed on the cut surfaces. The hatched thrips’ larvae notably concentrated and fed on these cut surfaces, which could be attributed to the fact that the exposed internal tissues of the pod offered a more abundant source of juice or scent, thereby attracting thrips for feeding. In the field, thrips typically feed on the tender tissues of cowpeas or cowpea flowers and will also consume cowpea pods [13]. However, in their natural habitat, thrips do not have cut surfaces of cowpeas. At present, there is no empirical evidence whether the standard rearing protocol of using cut food material has any impact on the survival and reproduction of thrips.

Since *M. usitatus* and *F. intonsa* share a common ecological niche, understanding their growth, development, and reproduction on different host plants may provide valuable insights into their competitive mechanisms. We developed age-stage, two-sex life tables for both species of thrips on three different bean pods, exploring their adaptability to different host plants so as to reveal the competition advantages of both species. As previous life table data of thrips have mostly been collected by feeding them on cut pods [10,11,12], we compared performance on cut and cut surface-covered pods, providing a new approach for studying thrips’ growth and development in a laboratory, which may better capture how thrips feed in the field.

## 2. Materials and Methods

### 2.1. Insects and Plants 

Insects *(M. usitatus* and *F. intonsa*) were sourced from field cowpea planted in the Dafeng base (Guangdong Academy of Agricultural Sciences, Plant Protection Institute). Both species of thrips were continuously maintained on snap bean pods for more than one year (over 10 generations) in separate constant-temperature rooms, 26 ± 1 °C; relative humidity at 75%; and 14L: 10D photoperiods [10]. 

The pod length of cowpea (*Vigna unguiculata* L. var. Chunqiu Wujiadou) is 45–60 cm, and the pod width is 0.9–1.2 cm; snap bean (*Phaseolus vulgaris* L. var. Yonglong 3) pods are straight and flat, with the pod width at about 3 cm and pod length at about 35–40 cm; green bean (*Phaseolus vulgaris* L. var. Linghangzhe) has round pods, with the pod width at about 1–1.5 cm and pod length at about 23–30 cm. The beans were grown in an insect-proof shed outside the laboratory.

### 2.2. Determination of Life Table Parameters of Two Species of Thrips on Three Types of Bean Pods

Cohorts of *M. usitatus* and *F. intonsa* were transferred to fresh cut pods of cowpea, snap bean, and green bean. After one generation, 25 newly emerged male and female adults were placed in glass jars and provisioned with fresh pods to lay eggs. Bean pods were replaced and checked every 24 h to obtain 100 newly hatched larvae, which were randomly selected and placed individually in rearing devices made by 10 mL centrifuge tubes; the bottom of the tube was cut off and covered with a 200-mesh gauze and the lid was removed and sealed with parafilm after thrips and pods were transferred. Larvae were fed with cut sections of pods replaced every three days. When the larvae entered the prepupal stage, they were transferred to 24-well plates with moist filter paper. Each well contained one prepupae and fresh pod, then a modified tube was inserted to cover each well, preventing thrips from escaping. The pods were regularly replaced until the adult thrips died. Observations were made every 24 h to record the developmental time and mortality of each stage, identifying each stage and sex based on published morphological characteristics [14], and longevity of adults was recorded. Since eggs were laid into plant tissue and not visible, the egg period was recorded from the end of oviposition to the appearance of larvae, but the number of eggs and egg survival rate could not be determined. 

### 2.3. Fecundity of Two Species of Thrips on Different Bean Pods

Female thrips from the above experiments were continually reared in 24-well plates with fresh cut bean pods. A newly emerged male was placed in each well for mating, and if the male died, it was replaced with a newly emerged male. The bean pods were replaced every 24 h until the females died. The replaced pods were placed in new modified tubes and sealed with parafilm. The number of larvae emerging from the pods was recorded until pods decayed (approximately 6 days). Fecundity was estimated based on the number of emerging larvae. 

### 2.4. Effect of Shrinkable Film on Survival of M. usitatus

We provided cut cowpea segments approximately 3 cm in length and sealed the middle of each segment with a shrink film approximately 1 cm long; untreated cowpea segments were used as control to explore effects of shrink film on the survival of adult thrips. Each treatment contained 6 replicates and each replicate included 20 thrips; cowpea segments were contained in a 9 cm Petri dish lined with round moistened filter paper. Petri dishes were sealed with parafilm to prevent thrips from escaping. Survival of adult thrips was recorded every 24 h for 7 days. 

### 2.5. Survival and Fecundity of Adult M. usitatus on Differently Treated Cowpea Segments

Two types of cowpea segments (approximately 1.5 cm long) were provided for adult thrips to feed and lay eggs: middle-covered (MC) and both end-covered (BC) using shrink film. A pair of newly emerged male and female adults were placed in a 3 cm Petri dish together with a treated bean pod, and a round moistened filter paper was placed under the pods to maintain humidity. Each treatment included 16 replicates and each replicate included two adults. Petri dishes were sealed with parafilm, and a wallpaper knife was used to create cracks on the lid of the Petri dish for ventilation. The bean pods were replaced every 24 h until the female adult died and the date of its death was recorded. The discarded ovipositing bean pods were placed in a new Petri dish and sealed. The number of hatched larvae were observed and recorded daily for approximately 6 days, until no newly hatched larvae appeared. The total number of hatched larvae on each bean pod was used as an estimate of oviposition.

### 2.6. Data Analysis

Life table data were statistically analyzed using the age-stage, two-sex life table theory with TWOSEX-MSChart software [15,16,17,18]. For each species of thrips, the calculated parameters included the survival rate (*l_x_*), fecundity (*m_x_*), net reproductive rate (*l_x_m_x_*), age-stage-specific survival rate (*s_xj_*, where *x* is age and *j* is stage), the age-stage-specific fecundity (*f_xj_*), life expectancy (*e_xj_*), and reproductive value (*v_xj_*); population parameters: the net reproduction rate (*R*_0_), intrinsic rate of increase (*r*), finite rate of increase (*λ*), mean generation time (*T*), and gross reproduction rate (*GRR*) were also calculated according to the equations and software mentioned in the article by Chi et al. [6]. Variance and standard errors of the demographic and population parameters were computed using paired bootstrap tests with 100,000 resampling iterations, and significance tests were conducted for differences [6], plotting with SigmaPlot 15.0.

The results of the survival bioassays were subjected to a survival analysis performed using the Kaplan–Meier estimators (log-rank method) with GraphPad Prism 8. The log-rank (Mantel–Cox) test was used to calculate the curve comparison. For the analysis of fecundity, a two-tailed unpaired *T* test was used.

## 3. Results

### 3.1. Development and Fecundity of Two Thrips on Three Bean Pods

The total preadult periods of both species of thrips were shortest on green bean pods. Additionally, female adult longevities of *M. usitatus* and *F. intonsa* were shortest on green bean pods, and their fecundities were also the lowest. There was no significant difference in female adult longevity of *M. usitatus* between cowpea pods and snap bean pods. However, *M. usitatus* exhibited higher fecundity on snap bean pods. For *F. intonsa*, female adult longevity and fecundity were higher on cowpea pods (Table 1).

On cowpea pods, there was no significant difference in adult longevity between these two thrips, but female *F. intonsa* had a significantly longer oviposition period and higher fecundity than *M. usitatus*. On snap bean pods, there was no significant difference in both the male longevity and female oviposition period between the two species, but female *M. usitatus* had longer longevity and higher fecundity. On green bean pods, there was no significant difference in female longevity between the two species, but male *F. intonsa* lived longer, and females had a longer oviposition period and higher fecundity than *M. usitatus* (Table 1).

### 3.2. Age-Stage and Age-Specific Survival Rate and Fecundity of Two Thrips on Three Bean Pods

The age-stage specific survival rate (*s_xj_*) represents the probability of newly hatched thrips surviving to a specific age *x* and stage *j* (Figure 1). There was a significant overlap in the curves, indicating variation in the individual development rate of thrips. The survival rates of female adult thrips on different bean pods were higher than those of males. Female thrips of both species had the longest longevity on snap bean pods (*M. usitatus*: 40 days, *F. intonsa*: 37 days). The larval and pupal period of both species exhibited the highest survival rates on cowpea pods, while the lowest survival rates in the pupal period and the shortest adult longevity were observed on green bean pods. Among these three types of bean pods, female longevity of *M. usitatus* was consistently longer than that of *F. intonsa*.

All *l_x_* curves remained flat during the egg period, decreased during the larval period, flattened again during the pupal period and initial adult stage, and decreased during adulthood (Figure 2). This pattern indicates low mortality rates during the pupal period and higher mortality rates during the larval and late adult period for both species of thrips under laboratory conditions. The *f_x_*_5_ (the age-stage-specific fecundity, *j* = 5 represents the fifth stage: egg, larva, prepupa, pupa, and female), *m_x_*, and *l_x_m_x_* values of all treatments exhibited an initial increase followed by a decrease, with two distinct peaks. However, occurrence time of these peaks varied among hosts and between thrips. For *M. usitatus*, the highest values of *f_x_*_5_, *m_x_*, and *l_x_m_x_* appeared at day 18 on cowpea pods and at day 23 on both snap bean and green bean pods (Figure 2), while for *F. intonsa,* the highest values of *f_x_*_5_, *m_x_*, and *l_x_m_x_* appeared at day 18 on cowpea and green bean pods and at day 22 on snap bean pods (Figure 2).

### 3.3. Life Expectancy and Reproductive Value of Two Thrips on Three Bean Pods

The age-specific life expectancy (*e_xj_*) indicates the expected longevity of individual thrips feeding on different bean pods. The expected longevity of offspring on cowpea pods, snap bean pods, and green bean pods was 22.9 d, 20.9 d, and 14.1 d for *M. usitatus* and 23.9 d, 19.2 d, and 13.8 d for *F. intonsa*. The expected longevity on cowpea was 21.9 d for female *M. usitatus* and 21 d for *F. intonsa*, while it was 22.5 d for *M. usitatus* and 19.7 d for *F. intonsa* on snap bean pods (Figure 3).

The age-specific reproductive values (*v_xj_*) represent the average contribution of an individual of age *x* and stage *j* to the future population development when feeding on different bean pods. The reproductive values of both species increased with age until reaching a peak, followed by a decline. However, the peak and rate of decline varied depending on the bean pod type. The peak reproductive values of *M. usitatus* were, respectively, observed at 15 d (32.85 day^−1^) on cowpea, 16 d (49.59 day^−1^) on snap bean, and 14 d (24.63 day^−1^) on green bean; and the peak reproductive values of *F. intonsa* were observed at 13 d (43.99 day^−1^) on cowpea, 15 d (39.55 day^−1^) on snap bean, and 12 d (26.48 day^−1^) on green bean (Figure 4).

### 3.4. Life Table Parameters of Two Thrips on Three Bean Pods

Population parameters describe the growth potential of a population, and they reflect the overall impact of survival, and reproductive ability on population fitness. Feeding on different bean pods affected the population parameters of both species (Table 2). Both species exhibited lower population parameters when feeding on green bean. *M. usitatus* showed the highest net reproductive rate (*R*_0_), finite rate of increase (*λ*), intrinsic rate of increase (*r*), and gross reproduction rate (*GRR*) when feeding on snap bean. *F. intonsa* showed the highest *Ro*, *λ*, and *r* when feeding on cowpea. On cowpea pods, *F. intonsa* had higher *R*_0_, *r*, *λ*, and *GRR* than *M. usitatus*, while mean generation time (*T*) of *F. intonsa* was shorter than that of *M. usitatus*. On snap bean, *M. usitatus* had higher *R*_0_, *GRR*, and *T* than *F. intonsa*, but *r* and *λ* were of no significant difference between *M. usitatus* and *F. intonsa.*

### 3.5. Survival and Fecundity of M. usitatus on Different Treated Cowpea Segments

There was no significant disparity in the survival rate between *M. usitatus* fed on untreated cowpea segments and segments wrapped with shrink film in the middle (Figure 5a). This suggested that the introduction of shrink film in subsequent experiments had little influence on the survival of thrips.

Thrips fed on cowpea segments sealed at both ends exhibited significantly reduced fecundity compared to those fed on middle-wrapped segments (Figure 5b). The age-dependent fecundity pattern for thrips (Figure 5(c1,c2)) feeding on both types of treated cowpeas peaked shortly after the commencement of oviposition, followed by fluctuations, and a decline in the late stage of female adulthood. There were three peaks in the curves of daily fecundity. For thrips feeding on middle-wrapped cowpea segments, three peaks were, respectively, observed on day 7, 11, and 14, with peak values of 9.06, 7.00, and 7.8. Meanwhile, for end-wrapped cowpea, three peaks appeared at day 8, 14, and 18, with peak values of 7.25, 4.75, and 3.94, which also indicated a reduction in fecundity.

## 4. Discussion

Thrips like *M. usitatus* and *F. intonsa* are major pests of cowpea in South China. Due to their short generation cycle, high fecundity, and overlapping generations, they pose a significant threat to the production of cowpea [19,20]. Interspecific competition is considered a common phenomenon among almost all taxonomic units, especially in Insecta [21,22]. In this study, *F. intonsa* developed faster on cowpea pods compared to *M. usitatus*, and its fecundity was significantly higher. Compared to *M. usitatus*, population parameters of *F. intonsa*, including *R*_0_, *r*, *λ*, and *GRR*, were all higher on cowpea but with a lower T, which suggested that *F. intonsa* had a competitive advantage on cowpea, consistent with previous studies [12]. However, the GRRs were much higher in study [12]; we assume that it was because its experiments were carried out in an outdoor net room, which provided a field environment for thrips to live, perhaps contributing to a higher reproduction rate. Moreover, field surveys indicate that *M. usitatus* is gradually replacing *F. intonsa* as the dominant species on cowpea, which might be related to insecticide use in the field, such as spinetoram [5]. The differences in resistance to different insecticides between these two species of thrips and related mechanisms of resistance require further monitoring and study. 

Previous studies on *M. usitatus* mainly used cowpea as a food source, with less research using other leguminous plants [12,23,24]. In this study, age-stage, two-sex life tables of *M. usitatus* were generated on two more bean varieties, snap bean and green bean. The population parameters of *M. usitatus* on snap bean, including *R*_0_, *r*, *λ*, and *GRR*, were all higher than on other bean pods, while the T value was the highest. The oviposition period of *M. usitatus* on snap bean was the longest, and fecundity was the highest, indicating that a snap bean pod is a more suitable source of food, which was consistent with previous research [8]. However, field reports of *M. usitatus* damage on cowpea are more common. In South China like Guangxi, Guangdong, and Hainan, *M. usitatus* has caused severe damage to cowpea [23], while reports of its damage to snap bean are far less serious. This may be related to climate conditions and the geographical distribution of bean production. Annual average temperature in South China is around 20–25 °C, which is suitable for the growth and development of both *M. usitatus* and cowpea [25,26]. In addition, unlike snap bean, cowpea is an important economic crop in South China and extensively cultivated year-round, especially during cold seasons to supply the north market. Meanwhile, snap bean is mainly grown in the north or southwest of China (Henan, Hebei, Shandong, and Sichuan Provinces) where climate is less suitable for *M. usitatus* [27]. Therefore, reasonable adjustments of planting seasons and the local arrangement of bean varieties should be integrated into the IPM strategies for thrips. 

A shorter preadult period often points to a better host plant while low fecundity indicates the opposite [11,12]. However, in our study, the total preadult periods of the two species of thrips on green bean were the shortest, but the female adult longevity and fecundity were the lowest (Table 1), which suggested that green bean is a “two-faced” host for both species of thrips. We speculated that it might be related to the physiological traits of green bean. The epidermis of green bean is hard and with fine hairs, which challenges female thrips to lay eggs into plant tissue. Meanwhile, with their piercing–sucking feeding mode [1], thrips can successfully steal the nutrients from green bean, which allows a short preadult period to appear. But the specific reasons need to be further studied.

In the field, *M. usitatus* predominantly relies on intact flowers, tender leaves, and young pods of cowpea [28], while segments of cowpea pods were used as food supply in laboratory experiments. Once the cowpea is segmented, juicy cut surfaces are exposed to thrips, which could influence the parameters of thrips’ development. We investigated whether direct access to cut surfaces influenced the survival and fecundity of *M. usitatus.* Results showed that thrips with direct access to cut surfaces exhibited higher fecundity, while there was no significant effect on survival (Figure 5). It is hypothesized that the juicy cut surfaces could possibly enhance the oviposition of female *M. usitatus* and also the hatching of eggs (since hatched larvae indicate fecundity). Further investigations are ongoing to explore this issue in more detail. Feeding and oviposition conditions without direct access to surfaces of cowpea pods resemble field conditions more closely, making data more pertinent for analyzing field populations. Avoiding direct contact of thrips to cut surfaces may help improve methods for life table studies of thrips.

## 5. Conclusions

The life table is an essential tool for studying insect population dynamics. In this study, impacts of different host plant species (varieties) on population dynamics of two species of thrips were evaluated by life tables, providing primary references for selecting insect-resistant varieties [29,30]. In conclusion, this study generated age-stage, two-sex life tables of *M. usitatus* and *F. intonsa* on three bean varieties, suggesting a competitive advantage of *M. usitatus* on snap bean while *F. intonsa* has the advantage on cowpea. We found that cut surfaces of pods used in regular life table studies of thrips influenced their development parameters, which suggested that end-wrapped cut surfaces could be a better option as a source of food for life table analyses. Our study helps to provide data for understanding the competitive mechanism between two major pests, thrips, on bean varieties and is the first to highlight the influence of cut surfaces on parameters of life tables.

## Figures and Tables

**Figure 1 insects-15-01003-f001:**
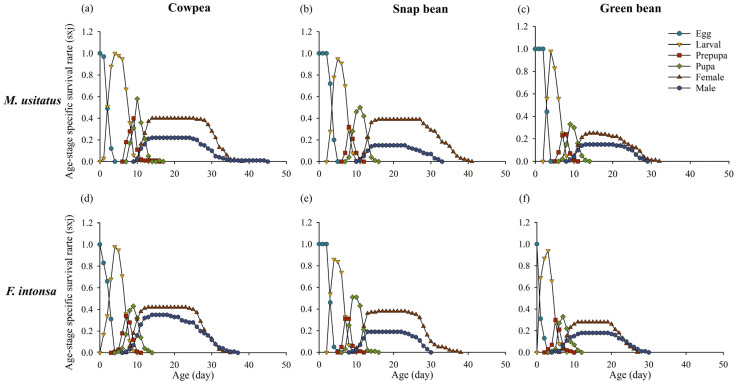
Age-stage-specific survival rate (sxj) of *Megalurothrips usitatus* and *Frankliniella intonsa* on three bean pods: (**a**) *M. usitatus* on cowpea, (**b**) *M. usitatus* on snap bean, (**c**) *M. usitatus* on green bean, (**d**) *F. intonsa* on cowpea, (**e**) *F. intonsa* on snap bean, and (**f**) *F. intonsa* on green bean.

**Figure 2 insects-15-01003-f002:**
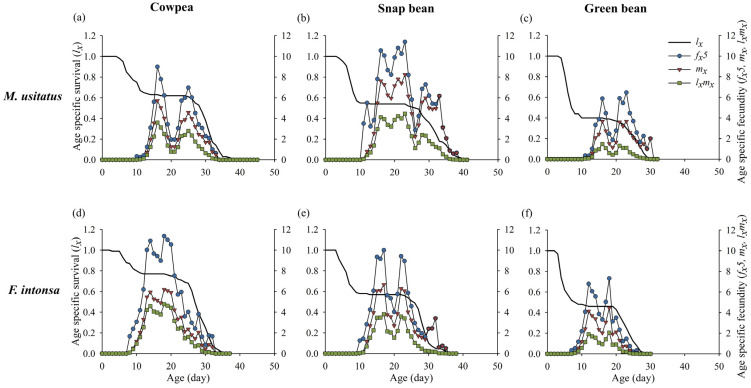
The age-specific survival rate (*l_x_*) and fecundity (*f_x_*_5_, *m_x_*, *l_x_m_x_*) of *Megalurothrips usitatus* and *Frankliniella intonsa* on three bean pods: (**a**) *M. usitatus* on cowpea, (**b**) *M. usitatus* on snap bean, (**c**) *M. usitatus* on green bean, (**d**) *F. intonsa* on cowpea, (**e**) *F. intonsa* on snap bean, and (**f**) *F. intonsa* on green bean. Note: *f_xj_* is the age-stage-specific fecundity. Here, *j* represents the fifth stage (egg, larvae, prepupa, pupa, female) of both species of thrips, the female stage.

**Figure 3 insects-15-01003-f003:**
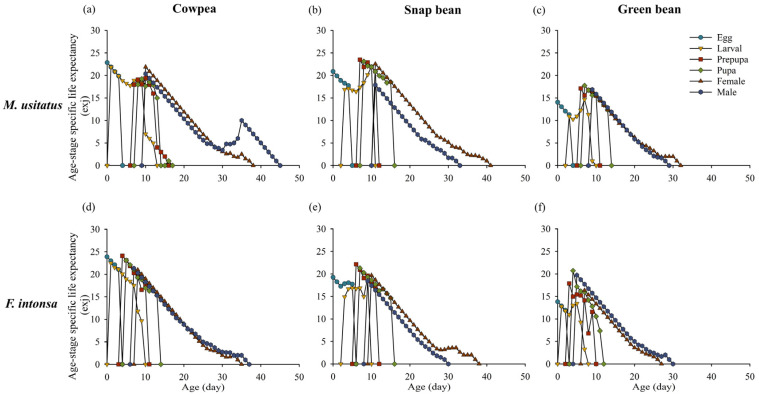
Age-stage-specific life expectancy (*e_xj_*) of *Megalurothrips usitatus* and *Frankliniella intonsa* on three bean pods: (**a**) *M. usitatus* on cowpea, (**b**) *M. usitatus* on snap bean, (**c**) *M. usitatus* on green bean, (**d**) *F. intonsa* on cowpea, (**e**) *F. intonsa* on snap bean, and (**f**) *F. intonsa* on green bean.

**Figure 4 insects-15-01003-f004:**
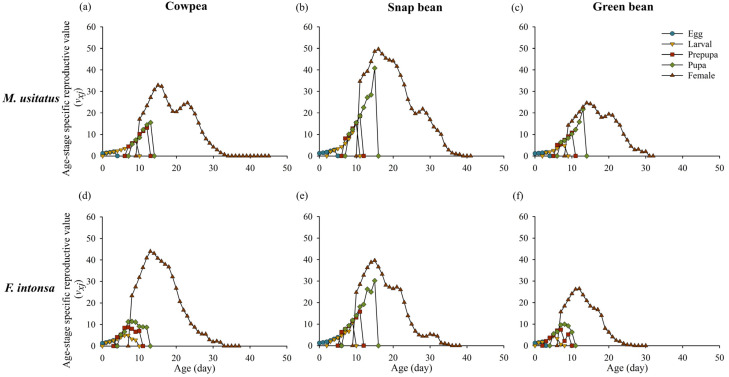
Age-stage-specific reproductive value (*v_xj_*) of *Megalurothrips usitatus* and *Frankliniella intonsa* on three bean pods: (**a**) *M. usitatus* on cowpea, (**b**) *M. usitatus* on snap bean, (**c**) *M. usitatus* on green bean, (**d**) *F. intonsa* on cowpea, (**e**) *F. intonsa* on snap bean, and (**f**) *F. intonsa* on green bean.

**Figure 5 insects-15-01003-f005:**
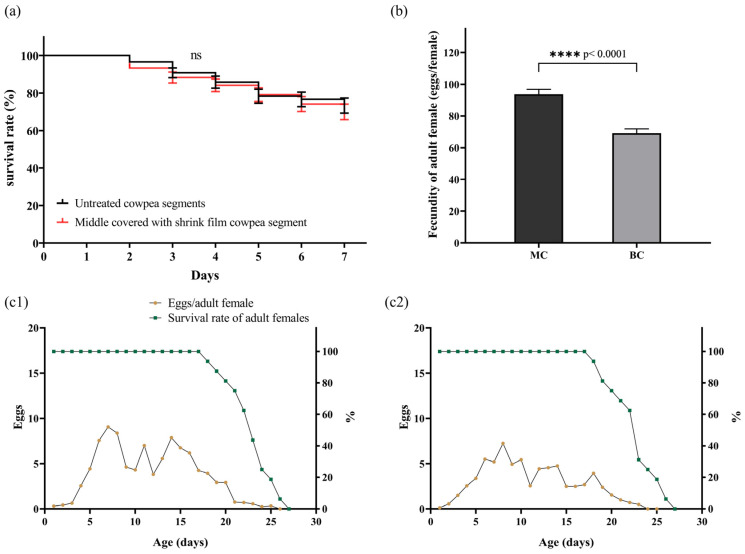
Survival and fecundity of *Megalurothrips usitatus* on different treatments of cowpea segments. (**a**) Kaplan-Meier of *M. usitatus* on different treated cowpea segments. (**b**) Fecundity of *M. usitatus* on different treatments of cowpea segments, MC: middle-covered cowpea segments, BC: both ends-covered cowpea segments. (**c1**,**c2**) Fecundity and survival of adult females of *M. usitatus* on different types of cowpea segments. (**c1**): MC, (**c2**): BC. Differences between means ± SEM in (**a**,**b**) were analyzed using two-tailed unpaired *t*-tests; ns, not significant.

**Table 1 insects-15-01003-t001:** Demographic parameters (means ± SE) of *Megalurothrips usitatus* and *Frankliniella intonsa* reared on three bean pods.

	*M. usitatus*						*F. intonsa*					
	*n*	Cowpea	*n*	Snap Bean	*n*	Green Bean	*n*	Cowpea	*n*	Snap Bean	*n*	Green Bean
Matched cohort size (female:male)		40:22		39:15		25:15		42:35		38:19		28:18
Initial cohort size (female:male)		25:25		25:25		25:25		25:25		25:25		25:25
Egg period (days)	100	2.58 ± 0.074 c*	100	3.92 ± 0.070 a*	100	3.44 ± 0.052 b*	100	2.80 ± 0.11 B	100	3.52 ± 0.064 A	100	1.44 ± 0.071 C
Larval period (days)	81	5.83 ± 0.095 a*	58	4.41 ± 0.12 b*	45	3.95 ± 0.11 c	83	4.34 ± 0.086 A	61	3.97 ± 0.11 B	58	3.84 ± 0.10 B
Prepupal period (days)	77	1.29 ± 0.061 a	56	1.21 ± 0.055 a	42	1.40 ± 0.084 a*	82	1.16 ± 0.041 A	59	1.24 ± 0.056 A	51	1.16 ± 0.052 A
Pupa period (days)	62	2.34 ± 0.095 b*	54	3.57 ± 0.10 a	40	2.50 ± 0.12 b*	77	1.86 ± 0.10 B	57	3.44 ± 0.14 A	46	2.02 ± 0.11 B
Total preadult period (days)	62	11.85 ± 0.14 b*	54	13.20 ± 0.15 a*	40	11.27 ± 0.18 c*	77	10.18 ± 0.17 B	57	12.05 ± 0.16 A	46	8.52 ± 0.20 C
Female adult longevity (days)	40	19.92 ± 0.39 a	39	20.26 ± 0.55 a*	25	13.95 ± 0.69 b	42	19.19 ± 0.40 A	38	17.48 ± 0.39 B	28	14.75 ± 0.41 C
Male adult longevity (days)	22	18.68 ± 0.88 a	15	15.87 ± 0.87 b	15	14.80 ± 0.46 b*	35	17.69 ± 0.78 A	19	15.63 ± 0.33 B	18	16.33 ± 0.63 AB
Oviposition period (days)	40	16.17 ± 0.26 a*	39	16.56 ± 0.47 a	25	10.20 ± 0.60 b*	42	17.97 ± 0.39 A	38	15.53 ± 0.39 B	28	11.97 ± 0.44 C
Fecundity (F) (eggs/female)	40	81.71 ± 2.36 b*	39	144.39 ± 7.49 a*	25	46.22 ± 2.90 c*	42	129.32 ± 4.17 A	38	97.83 ± 2.50 B	28	55.25 ± 2.78 C

A paired bootstrap test (B = 100,000) based on the confidence interval of differences between treatments was used for comparison. Lowercase lettering indicates significant differences between different bean varieties within *M. usitatus*, and uppercase lettering indicate significant differences between *F. intonsa* on different bean varieties. ‘*’ indicates significant difference between two thrips species on the same pod varieties (*p* < 0.05).

**Table 2 insects-15-01003-t002:** Population parameters (means ± SE) of *Megalurothrips usitatus* and *Frankliniella intonsa* reared on three bean pods.

		*R* _0_	*r*	*λ*	*T*	*GRR*
		Net Reproduction Rate	Intrinsic Rate of Increase	Finite Rate of Increase	Mean Generation Time	Gross Reproduction Rate
*Megalurothrips usitatus* (Bagnall)	cowpea	32.69 ± 4.11 b*	0.17 ± 0.01 b*	1.19 ± 0.01 b*	20.34 ± 0.22 a*	56.61 ± 5.31 b
	snap bean	56.32 ± 7.63 a*	0.19 ± 0.01 a	1.21 ± 0.01 a	20.71 ± 0.30 a*	124.08 ± 10.20 a*
	green bean	11.55 ± 2.13 c	0.12 ± 0.01 c*	1.13 ± 0.01 c*	19.46 ± 0.34 b*	37.83 ± 6.07 c
*Frankliniella intonsa* (Trybom)	cowpea	54.30 ± 6.61 A	0.24 ± 0.01 A	1.27 ± 0.01 A	16.86 ± 0.22 B	75.65 ± 8.22 A
	snap bean	37.17 ± 4.83 B	0.19 ± 0.01 B	1.21 ± 0.01 B	18.88 ± 0.28 A	77.03 ± 7.09 A
	green bean	15.46 ± 2.59 C	0.18 ± 0.01 B	1.19 ± 0.02 B	15.31 ± 0.23 C	34.82 ± 4.53 B

A paired bootstrap test (B = 100,000) based on the confidence interval of differences between treatments was used for comparison. Lowercase lettering indicates significant differences between different bean varieties within *M. usitatus*, and uppercase lettering indicate significant differences between *F. intonsa* on different bean varieties. ‘*’ indicates significant differences between two species of thrips on the same bean varieties (*p* < 0.05).

## Data Availability

The original contributions presented in this study are included in the article; further inquiries can be directed to the corresponding authors.

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
