# Peer review of "Age-Stage, Two-Sex Life Tables of Megalurothrips usitatus (Bagnall) and Frankliniella intonsa (Trybom) on Different Bean Pods Under Laboratory Conditions: Implications for Their Competitive Interactions"

_insects, 2024, doi:10.3390/insects15121003_

Round 1
Reviewer 1 Report
Comments and Suggestions for Authors
My comments and suggestions are presented in the attached document (review report_KDK (bírálat).docx).

Reviewer 2 Report
Comments and Suggestions for Authors
The manuscript entitled "The life table parameters of Megalurothrips usitatus (Bagnall) and Frankliniella intonsa (Trybom) on different bean pods under laboratory conditions: Implications for their competitive interactions" requires major revision before it can be considered for publication.
-
The introduction needs to be strengthened by including specific data on the percentage of damage caused by thrips worldwide and particularly in China on the three types of beans mentioned in the study. This will provide a clearer context and underscore the significance of your research.
-
The methodology section requires more detailed information, particularly regarding the procedure for recording data for the two-sex life table analysis. Additionally, the statistical analyses used need to be justified thoroughly. Please ensure that all steps and parameters are described explicitly to allow replication of the study.
-
The quality of the figures needs improvement to enhance readability and clarity. Ensure that all graphical representations are of high resolution and well-labeled.
-
Reorganize the discussion section to ensure a logical flow of ideas. The discussion should systematically compare the life table parameters and explain their implications for the competitive interactions between the two thrips species on the different bean pods. Avoid redundancy and ensure each point adds value to the interpretation of results.
-
Rewrite the conclusion to be concise and impactful, summarizing the key findings and their broader implications. The conclusion should align with the research objectives and offer clear takeaways for future research or practical applications.
-
The manuscript needs to be carefully proofread to correct grammatical errors and improve sentence structure throughout the text.
- Please check the annotated pdf file.

Need improvement
Reviewer 3 Report
Comments and Suggestions for Authors
The manuscript by Mengni et al. deals with the life table parameters of the major cowpea pests, Megalurothrips usitatus and Frankliniella intonsa, on cowpea, snap bean, and green bean. M. usitatus exhibited higher fecundity on snap bean, while F. intonsa demonstrated better population growth on cowpea. F. intonsa outcompeted M. usitatus on cowpea, showing higher population parameters. Additionally, the survival of M. usitatus was unaffected by feeding directly on cut surfaces of cowpea, but its fecundity decreased when feeding was not allowed, suggesting the need for further research on thrips growth in laboratory conditions. Although the manuscript contains useful and interesting data, I have several major comments and miscellaneous suggestions for the authors to consider, which are itemized below.
Major points:
- The authors claim in Table 2 that a multiple comparison test using a paired bootstrap method was provided for life table parameters; however, no such comparisons appear in the table. It would be helpful if the authors could ensure these comparisons are clearly presented, as stated.
- The TWOSEX-MSChart tool is not only for life table analysis but also enables the analysis of biological parameters (e.g., longevity, duration, oviposition days, etc.) after bootstrapping. However, the authors used a one-way ANOVA without considering a two-way ANOVA, which could allow them to analyze the species, host plants, and their interactions as fixed factors. For consistency, I suggest that the authors consider using TWOSEX-MSChart for the biological parameter comparisons presented in Table 1.
- It would strengthen the manuscript if the authors could provide more details on earlier studies directly related to this research, specifically emphasizing the differences and advancements of their own study in comparison. I provide the reference numbers of these similar earlier studies for the authors’ consideration.
- There are a few references in the manuscript that may not be easily accessible or might not be the most relevant for your study. I have provided further details below.
Miscellanous:
L32, 33, 34: It is generally better not to use numerical expressions in the abstract. I suggest synthesizing the results without providing numerical values, which can make the abstract more concise and engaging for the readers.
L45-46: This statement seems incomplete and could be more informative. Suggested revision: "Our study suggests that experiments involving cut surfaces may be misleading, and further investigations are ongoing to address these concerns."
L51: Reference number 1 may not be directly related to the hosts of thrips. While this reference appears to be less accessible online, it is a systematic study of Thrips species in China. I recommend citing a review paper or a book on the host range of thrips instead, as this would more directly relate to the focus of your research.
L82: I suggest the authors provide more details on these earlier studies, specifically highlighting how the current study differs from references 9 and 10. This would give the reader a clearer understanding of the unique contributions of this work.
Table 1, L184: For consistency, biological and fecundity data should be analyzed using TWOSEX-MSChart with a paired bootstrap test, as previously mentioned.
L201: Can the authors clarify what "fx5" refers to in this context? It would be helpful to ensure all abbreviations and terms are defined clearly for the reader.
L211-212: Do you have any statistical tests or comparisons for the longevity of thrips species across host plants? I suggest avoiding comparative statements unless they are supported by statistical analysis, as numerical differences alone do not necessarily indicate statistical significance.
Table 2, L240: The paired bootstrap grouping letters appear to be missing in Table 2. Incorporating these letters would allow for clearer interpretation of the statistical comparisons.
Figure 5a: A t-test is not the most suitable method for comparing survival rates. I suggest using a more appropriate method, such as Kaplan-Meier, to analyze survival data.
L283: Please indicate that these regions are located in the southern part of China to help readers understand the geographic context of the study.
L290: It would be helpful to clarify whether these four regions are located in the south, southeast, or west of China. Providing this information will improve the clarity of the geographical context.
L312-313: This sentence would benefit from a more formal tone. I recommend removing the exclamation mark and rewriting it as: "Further investigations are ongoing to explore this issue in more detail."
References 32 and 33: References 32 and 33 may not be the best match for the focus of your study. It would be beneficial to consider citing papers that are more closely aligned with your specific research questions. For example, reference 32 focuses on host plant species rather than varieties. Since your study involves testing different plant species, you might want to explore studies that address this aspect. Research on varieties, linesor landraces with different morphological or biological features (such as earliness, leaf thickness etc) could have important implications for pest management and the development of resistant cultivars.
References 20 and 21: Please verify if references 20 and 21, cited in L159, are relevant to your work. In reference 20, a completely different pseudoreplication methodology (jackknife) was used for life table analysis. The jackknife method provides very limited replication (n-1), which may not align with the bootstrap method used in your study. I suggest verifying the appropriateness of these references and considering citing studies that employed the bootstrap method for pseudoreplications, which would be more relevant to your work.
Round 2
Reviewer 1 Report
Comments and Suggestions for Authors
The manuscript have been improved a lot. I only have a few short comments left.
First, in Line 102-105 I think it was a good idea to give more detail about the size and shape of bean pods, but I would suggest to add these data in a more uniform way. Length of bean pods are given for all three types of pods correctly, however, for the other dimension "thickness" is given for cowpea, "pod width" for snap bean, but nothing exact (only the fact that they "have round pods") for green bean.
Second, in Line 146 I think that the "BC" and "MC" abbreviations were mixed and put after the wrong phrases.
Third, I would still advise to give more details about how the calculation of the population parameters were made which are showed in Table 2, since they are not mentioned in the Materials and Methods. If the authors think that it would be an unnecessarily long explanation, than perhaps to put an exact citation into the Materials and Methods chapter would also help.
Lastly, in the References I think there is a mistake; the reference in Line 367-368 should be deleted.
Reviewer 2 Report
Comments and Suggestions for Authors
Well revised.
Author Response
We are happy that our revised version meet the requirement for the journal and do appreciate for the reviwer giving us really good suggestions which have improved our manuscript a lot.
Reviewer 3 Report
Comments and Suggestions for Authors
Dear Editor,
I have carefully reviewed the manuscript by Mengni Li et al. Upon checking, I noticed that the authors have addressed my earlier comments and suggestions. They have incorporated the necessary changes into the text, modified the references, and responded to my comments appropriately.
I now believe the manuscript can be accepted, with a few minor corrections. For example, line 158 should read 'Data Analysis.' Additionally, in the Data Analysis section, the authors should specify that a paired bootstrap test was used for multiple comparisons in the demographic and life table parameters .
Best regards,
Author Response
We are happy that the manuscript meets the requirement and do appreciate for the reviewer giving us really good suggestions which have improved our manuscript a lot. We have also changed "analysis" to "Analysis" in line 158 and specified that a paired bootstrap test was used for multiple comparisons in the demographic and life table parameter. (See line 166-168)